# ClusterGen: Token Generation in Sublinear Time and Memory with Clustering KV Cache

## Abstract

Despite the significant success of large language models (LLMs), their extensive memory requirements pose challenges for deploying them in long-context token generation. The substantial memory footprint of LLM decoders arises from the necessity to store all previous tokens in the attention module, a requirement imposed by key-value (KV) caching. In this work, our focus is on developing an efficient compression technique for the KV cache. Empirical evidence indicates a significant clustering tendency within key embeddings in the attention module. Building on this key insight, we have devised a novel caching method with sublinear complexity, employing online clustering on key tokens and online $\ell_2$ sampling on values. The result is a provably accurate and efficient attention decoding algorithm, termed ClusterGen. Not only does this algorithm ensure a sublinear memory footprint and sublinear time complexity, but we also establish a tight error bound for our approach. Empirical evaluations on long-context question-answering tasks demonstrate that ClusterGen significantly outperforms existing and state-of-the-art KV cache compression methods in terms of performance and efficiency.

## 1 Introduction

Large Language Models (LLMs) (Achiam et al., 2023; Touvron et al., 2023) play a crucial role in various natural language processing applications, including dialog systems (Taori et al., 2023; Chiang et al., 2023), coding assistance (Chen et al., 2021; Roziere et al., 2023), and image/video generations from text (Radford et al., 2021; Ho et al., 2022). All of these models rely on the transformer architecture, with the attention mechanism serving as the key component.

To fully harness the capabilities of LLMs, they must demonstrate both efficiency and accuracy in generating long sequences. In practical applications, deploying LLMs to generate tokens in an autoregressive manner involves a sequential decoding process, where attention is dynamically applied to each newly generated token. This process effectively constructs the output sequence in a streaming manner, one token at a time. Therefore, as the sequence grows, the model has to produce contextually relevant and coherent content.

A common method for autoregressive attention decoding involves the use of key-value (KV) caching, where key and value pairs from *all* preceding tokens are cached and reused to prevent redundant computations. However, this approach faces memory constraints, particularly when handling long sequences. In particular, the memory requirements and runtime for generating each new token increase linearly with context size, posing a significant challenge for efficient processing of extensive sequences. This linear scaling directly impedes practical applicability in real-world scenarios, such as chat systems, where large contexts are often encountered.

In this work, we delve into the primary computational and memory bottleneck of token generation. We propose ClusterGen, a novel approach designed to significantly reduce the memory and runtime complexity of token generation, moving from conventional linear growth to sublinear scale. To summarize, our goal is to answer the following question:

*Can we approximate the attention output in decoding phase*
*in sublinear space/time complexity in context length?*

### 1.1 RELATED WORK

Recent studies have underscored the need for efficient token generation, particularly with the rise of long-range context datasets. Several recent works have developed efficient strategies for compressing the KV cache. Zhang et al. (2023) proposed a greedy-type eviction algorithm that dynamically keeps at most $k \ll n$ token embeddings based on the accumulated attention scores where they refer to the Heavy Hitter Oracle (H2O). Liu et al. (2023a) empirically observed that tokens with initially high attention scores tend to stay high during the future generation process. Motivated by this observation, the authors proposed a strategy that only keeps the most recent and pivotal tokens whose attention scores are higher than a threshold. Ge et al. (2023) proposed an adaptive method of KV cache compression which identifies the intrinsic structures of attention heads and uses them to determine the optimal compression policy. Xiao et al. (2023) observed that a simple eviction mechanism that keeps only first few and last few tokens does not degrade much the decoding quality. They additionally proposed a fine-tuning method to solve performance degradation from their method. Liu et al. (2023b) developed an algorithm that reduces the generation latency by exploiting contextual sparsity. In addition to algorithmic acceleration, there has also been a line of work optimizing hardware resource configurations (Sheng et al., 2023; Hong et al., 2023). However, to the best of our knowledge, none of these works have achieved an efficient method for KV cache with fully sublinear-time memory space.

On the lower bound side, achieving subquadratic amortized runtime for producing output embeddings for $n$ tokens in the worst-case instances is likely impossible without making assumptions about the input tokens (Alman & Song, 2023; Sarlos et al., 2023). Therefore, to achieve fast runtime, it is necessary to rely on certain assumptions about the input tokens.

### 1.2 STREAMING ATTENTION PROBLEM

Deployment of LLMs involves computing attention output in a streaming fashion, where at each phase of token generation a triplet of vectors is given. More precisely, at $i$-th token generation phase, three vectors $\boldsymbol{q}_i, \boldsymbol{k}_i, \boldsymbol{v}_i \in \mathbb{R}^d$ are streamed where they are called by query, key and value, respectively. Suppose that $n$ of tokens have been streamed so far either in prompt encoding or token generation phase. The objective of streaming attention decoding is to compute the following:

$$\mathrm{Attn}(\boldsymbol{q}_n, \boldsymbol{K}_n, \boldsymbol{V}_n) = \mathtt{softmax}(\boldsymbol{K}_n \cdot \boldsymbol{q}_n)^\top \cdot \boldsymbol{V}_n, \tag{1}$$

where $\boldsymbol{K}_n, \boldsymbol{V}_n \in \mathbb{R}^{n \times d}$ are matrices defined by stacking the keys and values in their respective rows:

$$\boldsymbol{K}_n := \begin{bmatrix} \boldsymbol{k}_1^\top \\ \boldsymbol{k}_2^\top \\ \vdots \\ \boldsymbol{k}_n^\top \end{bmatrix}, \quad \boldsymbol{V}_n := \begin{bmatrix} \boldsymbol{v}_1^\top \\ \boldsymbol{v}_2^\top \\ \vdots \\ \boldsymbol{v}_n^\top \end{bmatrix}. \tag{2}$$

The output $\mathrm{Attn}(\boldsymbol{q}_n, \boldsymbol{K}_n, \boldsymbol{V}_n)$ is then used for predicting the next token and its token embedding is applied to a transformer model and introduce a new stream pair $(\boldsymbol{q}_{n+1}, \boldsymbol{k}_{n+1}, \boldsymbol{v}_{n+1})$ is generated. However, storing these values and keys requires $O(nd)$ memory, posing a significant space complexity challenge for long-context models with large $n$.

### 1.3 OVERVIEW OF CONTRIBUTIONS

We propose CLUSTERGEN, an efficient method that accurately approximates the attention decoder's output in Eq. (1) while retaining only a small (sublinear) subset of keys and values in the cache. In particular, CLUSTERGEN computes an estimator $\boldsymbol{z}_n$ for $\mathrm{Attn}(\boldsymbol{q}_n, \boldsymbol{K}_n, \boldsymbol{V}_n)$ in sublinear time and memory such that the error is bounded as follows:

$$\|\boldsymbol{z}_n - \mathrm{Attn}(\boldsymbol{q}_n, \boldsymbol{K}_n, \boldsymbol{V}_n)\|_2 \leq \varepsilon \|\mathtt{softmax}(\boldsymbol{K}_n \cdot \boldsymbol{q}_n)\|_2 \|\boldsymbol{V}_n\|_{\mathrm{op}}, \tag{3}$$

where $\|\cdot\|_{\mathrm{op}}$ represents the matrix operator norm. This error bound is in line with the spectral errors studied in previous works (Zandieh et al., 2023; Han et al., 2024).

We begin by observing that $\mathrm{Attn}(\boldsymbol{q}_n, \boldsymbol{K}_n, \boldsymbol{V}_n)$ in Eq. (1) is the product of the softmax vector $\mathtt{softmax}(\boldsymbol{K}_n \cdot \boldsymbol{q}_n)$ and value matrix $\boldsymbol{V}_n$. This matrix-vector product can be approximated by subsampling only $O(\varepsilon^{-2} d \log n)$ key-value pairs according to the vector and matrix according to the

squared norms of value tokens. This can be implemented in a streaming setting using some variants of reservoir sampling.

The other major computational challenge is computing the partition function in the denominator of the softmax function, i.e., $\sum_{i \in [n]} \exp(\langle \boldsymbol{k}_i, \boldsymbol{q}_n \rangle)$. To solve this, we construct a data structure that can be stored in sublinear memory and efficiently approximate $\sum_{i \in [n]} \exp(\langle \boldsymbol{k}_i, \boldsymbol{q}_n \rangle)$ up to $1 \pm \varepsilon$ factor for any query $\boldsymbol{q}_n$. Our method assumes that the key tokens can be covered by a sublinear number of bounded diameter clusters. This assumption is indeed weaker than the one made in Han et al. (2024), which in the decoding setting translates to having key tokens belong to only one cluster with a bounded diameter, while our approach allows for any sublinear number of clusters. So, if the keys are composed of bounded diameter clusters then we only need a small number of uniformly sampled keys from each cluster to approximate the softmax normalizer efficiently and accurately. The central task is to find these clusters in a streaming setting, and we achieve this using an algorithm that is inspired by the streaming k-center algorithm of Charikar et al. (1997).

In Theorem 2.4 and Corollary 2.5 we demonstrate that if the keys can be clustered into some sublinear number $m = n^{1-\Omega(1)}$ of clusters with some bounded diameters, then CLUSTERGEN operates with sublinear $O\left(\varepsilon^{-2}md\right) = O\left(\varepsilon^{-2}dn^{1-\Omega(1)}\right)$ memory and runtime and its output satisfies the approximation guarantee in Eq. (3). In Section 4, we empirically compare CLUSTERGEN to other KV cache compression methods including the attention-score-based algorithm of Zhang et al. (2023) and the deterministic eviction policy from Xiao et al. (2023). Our results confirm that CLUSTERGEN outperforms these methods, particularly in question-answering tasks with various sequence lengths.

## 2 SUBLINEAR TIME AND MEMORY ALGORITHM

Our goal is to approximate the attention output in Eq. (1) with a space complexity that is sublinear in context length $n$. To achieve this objective, we aim to design the following data structure (DS) for efficiently approximating the streaming attention mechanism:

### 2.1 STREAMING ATTENTION DATA STRUCTURE

For every positive integer $n$ and every stream of token triplets $(\boldsymbol{q}_1, \boldsymbol{k}_1, \boldsymbol{v}_1), \ldots, (\boldsymbol{q}_n, \boldsymbol{k}_n, \boldsymbol{v}_n)$ where $\boldsymbol{q}_i, \boldsymbol{k}_i, \boldsymbol{v}_i \in \mathbb{R}^d$, we aim to construct an efficient DS with the following properties:

- The required memory space is sublinear in $n$, i.e., $o(n)$.

- Upon the arrival of a new triplet $(\boldsymbol{q}_{n+1}, \boldsymbol{k}_{n+1}, \boldsymbol{v}_{n+1})$ in the stream, the time complexity to update is sublinear in $n$, i.e., $o(n)$.

- Given such data structure, there exists an algorithm that outputs an estimator $\boldsymbol{z}_n \in \mathbb{R}^d$ in sublinear time $o(n)$ such that:

$$\left\| \boldsymbol{z}_n - \texttt{softmax}(\boldsymbol{K}_n \cdot \boldsymbol{q}_n)^\top \cdot \boldsymbol{V}_n \right\|_2 \leq \varepsilon \left\| \texttt{softmax}(\boldsymbol{K}_n \cdot \boldsymbol{q}_n) \right\|_2 \left\| \boldsymbol{V}_n \right\|_{\text{op}}. \tag{4}$$

In the rest of this section, our focus is on developing an algorithm to satisfy the above properties. Note that the attention output in Eq. (1), using the definition of softmax, is equivalent to the following expression:

$$\text{Attn}(\boldsymbol{q}_n, \boldsymbol{K}_n, \boldsymbol{V}_n) = \frac{\exp(\boldsymbol{K}_n \cdot \boldsymbol{q}_n)^\top \cdot \boldsymbol{V}_n}{\sum_{i \in [n]} \exp(\langle \boldsymbol{k}_i, \boldsymbol{q}_n \rangle)}.$$

Thus, to compute the attention output we need to calculate:

1. The matrix-vector product between $\boldsymbol{V}_n$ and $\exp(\boldsymbol{K}_n \cdot \boldsymbol{q}_n)$.
2. The partition function $\sum_{i \in [n]} \exp(\langle \boldsymbol{k}_i, \boldsymbol{q}_n \rangle)$.

Thus, our DS needs to efficiently approximate each of these two operations. The matrix-vector product $\exp(\boldsymbol{K}_n \cdot \boldsymbol{q}_n)^\top \cdot \boldsymbol{V}_n$ can be approximated efficiently using standard sampling-based techniques. Specifically, we make use of the row norm sampling approach (Drineas & Kannan, 2001; Cohen et al., 2016). When multiplying two matrices $\boldsymbol{A} \in \mathbb{R}^{m \times n}$ and $\boldsymbol{B} \in \mathbb{R}^{n \times p}$, we randomly sample an i.i.d. index $i \in [n]$ with probability proportional to the $\ell_2$ norm of the $i$-th row in $\boldsymbol{B}$. Then, we

estimate $\boldsymbol{A} \cdot \boldsymbol{B}$ by the average of the product between $i$-th column in $\boldsymbol{A}$ and $i$-th row in $\boldsymbol{B}$. With this approximation, we need only $O(\varepsilon^{-2} d \log n)$ samples to guarantee an $\varepsilon$ multiplicative error in spectral norm for $\exp(\boldsymbol{K}_n \cdot \boldsymbol{q}_n)^\top \cdot \boldsymbol{V}_n$. Luckily, it can be implemented in a streaming setting through a variant of reservoir sampling (Vitter, 1985).

The more challenging task is the sublinear-time approximation of the partition function $\sum_{i \in [n]} \exp(\langle \boldsymbol{k}_i, \boldsymbol{q}_n \rangle)$. We construct a DS for computing this under the assumption that the keys in the token stream are organized into $o(n)$ of clusters. To be more precise, we introduce the following notion of clusterability:

**Definition 2.1** (Clusterability). For a positive integer $m$ and a real-valued $\delta > 0$, a dataset of points $\boldsymbol{x}_1, \boldsymbol{x}_2, \ldots \boldsymbol{x}_n \in \mathbb{R}^d$ is considered $(m, \delta)$-*clusterable* if there exists a size-$m$ partition $\mathcal{C}_1, \mathcal{C}_2, \ldots \mathcal{C}_m \subseteq \{\boldsymbol{x}_i\}_{i=1}^n$ of the dataset satisfying the following conditions:

- $\mathcal{C}_i \cap \mathcal{C}_j = \emptyset$ for every $i \neq j$ and $\bigcup_{j=1}^m \mathcal{C}_j = \{\boldsymbol{x}_i\}_{i=1}^n$.

- for every $j \in [m]$ and every distinct pair $\boldsymbol{y}, \boldsymbol{z} \in \mathcal{C}_j$, $\|\boldsymbol{y} - \boldsymbol{z}\|_2 \leq \delta$.

We demonstrate that under the assumption that the stream of keys $\boldsymbol{k}_1, \boldsymbol{k}_2, \ldots \boldsymbol{k}_n$ is $(m, \delta)$-clusterable as defined in Definition 2.1, with the number of clusters scaling sublinearly in stream length ($m = o(n)$), it is possible to construct a DS with sublinear memory space. The procedure for this DS is presented in Algorithm 1 which we refer to as CLUSTERGEN.

To verify this in the practical settings, we plot key embeddings from open-source LLMs in Section 3.1 and observe that they are indeed well clusterable on their embedding space. This motivates us to utilize an efficient stream clustering algorithm on key embeddings. In the remainder of this section, we provide a detailed explanation for the execution of the algorithm while simultaneously analyzing it through a series of lemmas.

## 2.2 MATRIX PRODUCT DATA STRUCTURE

Here, we focus on the UPDATEMATRIXPRODUCT primitive and establish its correctness by introducing invariants that are maintained throughout the stream processing. This primitive maintains and updates a list of $s$ elements denoted by $\mathcal{M}$ in CLUSTERGEN (Algorithm 1). Initially, this list is filled with null values. After processing the first token tuple $(\boldsymbol{q}_1, \boldsymbol{k}_1, \boldsymbol{v}_1)$, this list is populated with $s$ copies of the first key and value $(\boldsymbol{k}_1, \boldsymbol{v}_1)$. The procedure UPDATEMATRIXPRODUCT performs a variant of reservoir sampling upon observing any new token in the stream. At any iteration $n$ of the stream, $\mathcal{M}$ is ensured to contain $s$ i.i.d. samples chosen at random from $(\boldsymbol{k}_1, \boldsymbol{v}_1), \ldots, (\boldsymbol{k}_n, \boldsymbol{v}_n)$ with probabilities proportional to $\|\boldsymbol{k}_i\|_2^2$. More precisely, the following invariants hold:

**Lemma 2.2** (Correctness of UPDATEMATRIXPRODUCT). *For any positive integer $s$, at any iteration $n$ of the stream in Algorithm 1 the following properties are maintained:*

1. *$\mu = \sum_{i \in [n]} \|\boldsymbol{v}_i\|_2^2$.*

2. *$\mathcal{M}$ is a list of $s$ i.i.d. samples from $\{(\boldsymbol{k}_1, \boldsymbol{v}_1), \ldots, (\boldsymbol{k}_n, \boldsymbol{v}_n)\}$ where the probability distribution for each element $j \in [s]$ and $i \in [n]$ is $\Pr[\mathcal{M}(j) = (\boldsymbol{k}_i, \boldsymbol{v}_i)] = \frac{\|\boldsymbol{v}_i\|_2^2}{\sum_{l \in [n]} \|\boldsymbol{v}_l\|_2^2}$.*

The proof of the above lemma is deferred to Appendix A.1.

## 2.3 SOFTMAX NORMALIZER (PARTITION FUNCTION) DS

Here we delve into a detailed discussion of the UPDATESOFTMAXNORMALIZER primitive. This primitive constructs and maintains a DS denoted by $\mathcal{D}$, enabling accurate approximation of the partition function in the softmax denominator for any query. A crucial requirement for the efficiency of this primitive is that the key tokens must be $(m, \delta)$-clusterable, as per Definition 2.1. Our algorithm locates and stores a subsampled representation of each cluster in $\mathcal{D}$ in a small memory. Particularly, to achieve sublinear memory complexity, instead of keeping all keys in each cluster which would require $O(n)$ memory space, we maintain only a random subset of $t$ samples from each cluster.

Initially, $\mathcal{D}$ is an empty set. As new tokens in the stream are processed, new clusters get added to this set. Each cluster is characterized by a representative point, which is the first key assigned to

---

**Algorithm 1** CLUSTERGEN: Sublinear Streaming Attention via Clustering

---

1: **inputs:** stream of token embeddings $(\boldsymbol{q}_n, \boldsymbol{k}_n, \boldsymbol{v}_n)$, parameter $\delta > 0$, positive integers $s, t$

2: Initialize $\mu \leftarrow 0, \mathcal{D} \leftarrow \emptyset, \mathcal{M} \leftarrow \left[ \texttt{null}, \overset{\times s}{\dots\dots} \right]$

3: **repeat**

4:     $\mathcal{D} \leftarrow$ UPDATESOFTMAXNORMALIZER$(\mathcal{D}, \delta, t, \boldsymbol{k}_n)$

5:     $\mathcal{M} \leftarrow$ UPDATEMATRIXPRODUCT$(\mathcal{M}, s, \mu, \boldsymbol{k}_n, \boldsymbol{v}_n)$

6:     $\mu \leftarrow \mu + \|\boldsymbol{v}_n\|_2^2$

7:     $\boldsymbol{z}_n \leftarrow$ QUERYSTREAMATTN$(\mathcal{D}, \mathcal{M}, s, t, \mu, \boldsymbol{q}_n)$

8:     $n \leftarrow n + 1$

9:     **output** $\boldsymbol{z}_n$

10: **until** Token stream ends

---

**Procedure** UPDATESOFTMAXNORMALIZER $(\mathcal{D}, \delta, t, \boldsymbol{k})$

11: Initialize $\mathcal{D} \leftarrow \{(\boldsymbol{x}_i, \mathcal{S}_i, n_i) : i \in [m]\}$ and $i^* \leftarrow \arg\min_{i \in [m]} \|\boldsymbol{x}_i - \boldsymbol{k}\|_2$

12: **if** $\|\boldsymbol{k} - \boldsymbol{x}_{i^*}\|_2 \leq \delta$ **then**

13:     $n_{i^*} \leftarrow n_{i^*} + 1$

14:     Suppose $\mathcal{S}_{i^*}$ is a list of $t$ vectors in $\mathbb{R}^d$

15:     **for** $j \in [t]$ **do**

16:         Flip a coin and with probability $p = \frac{1}{n_{i^*}}$, update the $j^{th}$ entry of $\mathcal{S}_{i^*}$ as $\mathcal{S}_{i^*}(j) \leftarrow \boldsymbol{k}$

17:     **end for**

18: **else**

19:     $\mathcal{S}' \leftarrow \left[ \boldsymbol{k}, \overset{\times t}{\dots\dots} \right]$ (contains $t$ copies of $\boldsymbol{k}$)

20:     $\mathcal{D} = \mathcal{D} \cup \{(\boldsymbol{k}, \mathcal{S}', 1)\}$

21: **end if**

22: **return** $\mathcal{D}$

---

**Procedure** UPDATEMATRIXPRODUCT $(\mathcal{M}, s, \mu, \boldsymbol{k}, \boldsymbol{v})$

23: Suppose $\mathcal{M}$ is a list of $s$ tuples of vectors in $\mathbb{R}^d$

24: **for** $i \in [s]$ **do**

25:     Flip a coin and with probability $p = \frac{\|\boldsymbol{v}\|_2^2}{\mu + \|\boldsymbol{v}\|_2^2}$, update the $i^{th}$ entry of $\mathcal{M}$ as $\mathcal{M}(i) \leftarrow (\boldsymbol{k}, \boldsymbol{v})$

26: **end for**

27: **return** $\mathcal{M}$

---

**Procedure** QUERYSTREAMATTN $(\mathcal{D}, \mathcal{M}, s, t, \mu, \boldsymbol{q})$

28: $\boldsymbol{z} \leftarrow \sum_{(\boldsymbol{k}, \boldsymbol{v}) \in \mathcal{M}} \frac{\mu}{s \cdot \|\boldsymbol{v}\|_2^2} \cdot \exp(\langle \boldsymbol{q}, \boldsymbol{k} \rangle) \cdot \boldsymbol{v}$

29: $\tau \leftarrow \sum_{(\boldsymbol{x}, \mathcal{S}, n') \in \mathcal{D}} \frac{n'}{t} \cdot \sum_{\boldsymbol{k} \in \mathcal{S}} \exp(\langle \boldsymbol{q}, \boldsymbol{k} \rangle)$

30: **return** $\boldsymbol{z}/\tau$

---

that cluster by our algorithm. Throughout stream processing, we compute the distance between the new key token and each existing cluster. Here the distance to an existing cluster is defined as the distance to the aforementioned representative of the cluster. If there is a cluster whose distance is less than $\delta$, then the token is assigned to the nearest cluster, and we update our random samples of keys from this cluster using reservoir sampling. If the distance from all existing clusters is more than $\delta$, we introduce a new cluster in $\mathcal{D}$, and the new key becomes the representative of this new cluster. At any point in the stream, this algorithm identifies at most $m$ clusters if the keys so far are $(m, \delta)$-clusterable. If $m$ grows sublinearly in the stream length $n$, the memory and update time of our algorithm will be sublinear as well. Formally, we prove that the following invariant holds:

**Lemma 2.3** (Correctness of UPDATESOFTMAXNORMALIZER). *For any $\delta > 0$, any positive integer $t$, at any iteration $n$ of the stream in Algorithm 1 the following properties are maintained. $\mathcal{D}$ is a set of $m$ items of the form $\mathcal{D} = \{(\boldsymbol{x}_i, \mathcal{S}_i, n_i) : i \in [m]\}$, where there exists a partition of keys into $m$ disjoint subsets $\mathcal{C}_1, \mathcal{C}_2, \ldots \mathcal{C}_m \subseteq \{\boldsymbol{k}_i\}_{i=1}^n$ satisfying $\bigcup_{j=1}^m \mathcal{C}_j = \{\boldsymbol{k}_i\}_{i=1}^n$ and $\mathcal{C}_i \cap \mathcal{C}_j = \emptyset$ for every $i \neq j$, such that for every $i \in [m]$:*

  *1. $\boldsymbol{x}_i \in \mathcal{C}_i$,*

  *2. $n_i = |\mathcal{C}_i|$,*

3. $\|\boldsymbol{x}_i - \boldsymbol{k}'\|_2 \leq \delta$ for every $\boldsymbol{k}' \in \mathcal{C}_i$,

4. $\|\boldsymbol{x}_i - \boldsymbol{x}_j\|_2 > \delta$ for every $i \neq j$,

5. $\mathcal{S}_i$ is a set of $t$ i.i.d. uniform samples from the set $\mathcal{C}_i$.

The proof of Lemma 2.3 can be found in Appendix A.2.

## 2.4 STREAMING ATTENTION: MAIN THEOREM

Now we are ready to analyze the end-to-end performance of CLUSTERGEN and prove the main theorem. We show that, given the data structures created throughout the stream and analyzed in Lemma 2.2 and Lemma 2.3, the primitive QUERYSTREAMATTN can efficiently output an accurate approximation to the streaming attention, satisfying Eq. (1).

Our analysis unfolds in two steps. First, we establish that the data structures created by UPDATE-SOFTMAXNORMALIZER and UPDATEMATRIXPRODUCT can be stored in small memory and updated very quickly if the sequence of keys is clusterable into a sublinear number of clusters. Then we show that the QUERYSTREAMATTN can use these data structures to produce an accurate attention output for any given query. Our main result is as follows:

**Theorem 2.4** (Efficiency and Correctness of Algorithm 1). *For any $\delta, r, \varepsilon > 0$, any positive integers $n, d$, and any sequence of tokens $(\boldsymbol{q}_1, \boldsymbol{k}_1, \boldsymbol{v}_1), (\boldsymbol{q}_2, \boldsymbol{k}_2, \boldsymbol{v}_2), \ldots (\boldsymbol{q}_n, \boldsymbol{k}_n, \boldsymbol{v}_n)$ where $\boldsymbol{q}_i, \boldsymbol{k}_i, \boldsymbol{v}_i \in \mathbb{R}^d$, suppose that the followings hold*

- $t = \Omega\left(\varepsilon^{-2} \cdot e^{2\delta \cdot r} \log n\right),$

- $s = \Omega(\varepsilon^{-2} \cdot d),$

- $\|\boldsymbol{q}_n\|_2 \leq r.$

*Then, CLUSTERGEN (Algorithm 1) at $n$-th step of the stream processing outputs a vector $\boldsymbol{z}_n \in \mathbb{R}^d$ that satisfies Eq. (1) with probability at least $0.99$. Furthermore, if the keys $\boldsymbol{k}_1, \boldsymbol{k}_2, \ldots \boldsymbol{k}_n$ are $(m, \delta)$-clusterable as per Definition 2.1, then both the total memory of the algorithm and its runtime during the $n$-th iteration is bounded by $O(d \cdot (mt + s))$.*

The proof of Theorem 2.4 can be found in Appendix A.3.

**Memory and Runtime.** First, note that the memory requirement for storing the list $\mathcal{M}$ in Algorithm 1 is $O(sd)$ because it contains $s$ pairs of $d$-dimensional vectors. Next, to bound the memory requirement for storing $\mathcal{D}$ we need to bound the size of this set which we denoted by $m'$. According to properties **(1)** and **(4)** in Lemma 2.3, for every $i \in [m']$ there exist $\boldsymbol{x}_i \in \{\boldsymbol{k}_1, \boldsymbol{k}_2, \ldots \boldsymbol{k}_n\}$ such that $\|\boldsymbol{x}_i - \boldsymbol{x}_j\|_2 > \delta$ for $i \neq j$. Given the assumption in the theorem statement that keys are $(m, \delta)$-clusterable, by the definition of clusterability in Definition 2.1 along with the pigeonhole principle, we must have $m' \leq m$. Therefore storing $\mathcal{D}$ will require $O(m'td) = O(mtd)$ because it is a set of $m'$ elements, and each element of this set is a list of $t$ vectors in dimension $d$.

Three major operations dominate the runtime of the $n$-th iteration. Firstly, executing UPDATESOFT-MAXNORMALIZER requires computing $m'$ distances in line 12 that takes $O(md)$ time. Additionally, the for loop in line 16 takes $O(td)$ time. Secondly, UPDATEMATRIXPRODUCT has a runtime bounded by $O(sd)$. Thirdly, running QUERYSTREAMATTN involves $O(sd)$ operations in line 26 and $O(m'td) = O(mtd)$ operations in line 27. As a result, the total runtime of Algorithm 1 in $n$-th iteration is $O(mtd + sd)$.

Theorem 2.4 demonstrates that if the keys can be clustered into some sublinear number $m = n^{1-\Omega(1)}$ of clusters with diameters at most $\delta$, and the queries have bounded $\ell_2$ norms of at most $r$ such that the product of the cluster diameter and maximum $\ell_2$ norm of queries is bounded by $\delta r = o(\log n)$, then Algorithm 1 operates with sublinear $O\left(\varepsilon^{-2} \cdot mdn^{o(1)}\right) = O\left(\varepsilon^{-2} \cdot dn^{1-\Omega(1)}\right)$ memory and runtime. We summarize this in the following corollary:

**Corollary 2.5.** *Suppose the preconditions of Theorem 2.4 hold. If the diameter of key token clusters $\delta$ and the maximum $\ell_2$ norm of queries $r$ satisfy $\delta r = o(\log n)$, then the total memory and runtime*

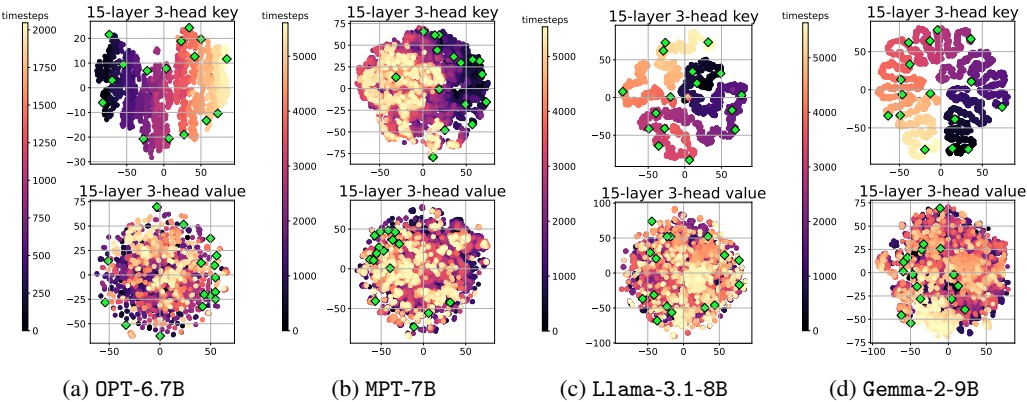

(a) `OPT-6.7B`     (b) `MPT-7B`     (c) `Llama-3.1-8B`     (d) `Gemma-2-9B`

Figure 1: A t-SNE plot of cached keys (first row) and values (second row) embeddings from 4 open-source models; (a) `OPT-6.7B`, (b) `MPT-7B`, (c) `Llama-3.1-8B` and (d) `Gemma-2-9B` using TriviaQA dataset. Key embeddings are more clusterable than value ones. The green dots represent the centers from the greedy k-center algorithm (Dyer & Frieze, 1985) where k=16.

| Model | Positional Encoding | Clusterability | Leaderboard Score (↑)[1] | Average $\|q_n\|_2$ |
|---|---|---|---|---|
| `OPT-6.7B` | AbsPE | Weak | Unknown | $15.753 \pm 0.0023$ |
| `MPT-7B` | ALiBi | Weak | 5.98 | $13.576 \pm 0.0012$ |
| `Llama-3.1-8B` | RoPE | Strong | 13.78 | $15.381 \pm 0.0005$ |
| `Gemma-2-9B` | RoPE | Strong | 20.98 | $23.800 \pm 0.0009$ |

Table 1: Language models with various positional encodings and their clusterability, performance scores from `open_llm_leaderboard` (Fourrier et al., 2024) (higher is better) and average $\ell_2$-norm of the query embeddings. The RoPE makes model performance strong and the corresponding key embeddings shows strong clusterabilities. All models have bounded query norms regardless of the sequence length.

of Theorem 2.4 are bounded by $O\left(\varepsilon^{-2} \cdot dmn^{o(1)}\right)$. Moreover, if the number of key token clusters $m$ grows as a sublinear function of $n$, i.e., as $m = n^{1-\Omega(1)}$, then the memory and runtime are bounded by $O\left(\varepsilon^{-2} \cdot dn^{1-\Omega(1)}\right)$.

The above results require that the key embeddings are well clusterable in their space, and the cluster centers should cover all keys with a small radius. In the next section, we empirically explore distributions on key and value embeddings and verify that the keys are indeed distributed in their embedding space. This supports that our assumption of clusterability on keys is reasonable in practical settings.

## 3 ABLATION STUDY

### 3.1 CLUSTERABILITY ON KEY AND VALUE EMBEDDINGS

We first demonstrate the clusterability of cached embeddings from long-range tokens. To this end, we collect key and value embeddings from 4 popular open-source language models; `OPT-6.7B`, `MPT-7B`, `Llama-3.1-8B` and `Gemma-2-9B`, where each adopts different positional encoding methods across absolute positional encoding (AbsPE), Attention with Linear Biases (ALiBi) (Press et al., 2021) and Rotational Positional Encodding (RoPE) (Su et al., 2024). We use prompts from TriviaQA dataset in LongBench (Li et al., 2023), and the length of input tokens is approximately 5,600 tokens, except for `OPT-6.7B`, which has a maximum sequence length of 2048. We then visualize the cached embeddings (at randomly selected layer/head) using t-SNE (Van der Maaten & Hinton, 2008), identifying cluster

---

[1] https://huggingface.co/spaces/open-llm-leaderboard/open_llm_leaderboard

center points through the greedy k-center algorithm (Dyer & Frieze, 1985). The benchmark models are summarized in Table 1, and their embeddings are illustrated in Fig. 1.

First, we observe that the key embeddings (first rows in Fig. 1) exhibit a higher degree of cluster-ability compared to value embeddings. Furthermore, we note that the cluster centers (indicated by green dots) corresponding to the key embeddings are evenly distributed across the entire embedding space. In particular, the key embeddings demonstrate significant dispersion across different time steps, and their cluster centers are distributed over the entire embedding space. Similar results are observed across various layers and heads and can be founded in Appendix B.1.

Second, it is clearly observed that the key embeddings in `Llama-3.1-8B` and `Gemma-2-9B`, both incor-porating RoPE, show strong clusterability. On the other hand, `MPT-7B` (using ALiBi) and `OPT-6.7B` (using AbsPE) demonstrate weaker clusterability. Notably, models utilizing RoPE have consistently outperformed those using other positional encoding methods across various benchmarks. For ex-ample, from `open_llm_leaderboard` (Fourrier et al., 2024), we observe that the performance ranking followed order of `Gemma-2-9B` > `Llama-3.1-8B` $\gg$ `OPT-30B` > `MPT-7B` (score for `OPT-6.7B` has not been submitted). These observations strongly support the validity and practicality of our clusterability assumption (Definition 2.1) for high-performing open-source LLMs.

### 3.2 Bounded Norm on Query Embeddings

We additionally investigate assumption on the upper bound of query embeddings in Theorem 2.4, i.e., $\|q_n\|_2 \leq r$ for some constant $r > 0$. Essentially, query embeddings are obtained by mul-tiplying weights by the input embeddings, and they are typically passed through the Layer Nor-malization (Ba, 2016). Therefore, entries of query embeddings are expected to be small assuming the weight matrices have small eigenvalues, and the norms of query embeddings in our case after Layer Normalization are expected to be small constants. We empirically compute $\ell_2$ norms of query embeddings from the same models used in Section 3.1 using the first 20 prompts from TriviaQA dataset. The average $\ell_2$-norms of the query (with 95% confidence interval) is reported in Table 1, which are mostly constant and upper-bounded regardless of the length of the input prompt. This strongly supports our assumption on upper bound of query embeddings.

## 4 Experiments

In this section, we report the empirical results of the proposed algorithm with memory footprint reduction and performance on various downstream question-answering tasks. For all experiments, we use a single NVIDIA A100 GPU with 80 GB VRAM .

### 4.1 Line Retrieval

We first evaluate our proposed algorithm on long-context line retrieval task in LongEval (Li et al., 2023)[2] benchmark. The task involves long-context line retrieval from extensive documents, each comprising multiple lines, complete with line numbers and topics. The objective is to precisely retrieve a specified number of lines corresponding to a target topic. We vary the number of lines, representing the number of targets, to 200, 300, and 400 and they correspond to sequence lengths of $n =$5,000, 7,000, and 9,000, respectively. Each dataset contains 50 distinct questions, and we systematically extract the number from the generated answers and compute accuracies. The answers are generated employing the `longchat-7B` model[3], which is a fine-tuned version of the Llama-2-7B model with long-range context length.

We compare our method to two KV cache compression algorithms; H2O (Zhang et al., 2023), which retains cached tokens with high cumulative attention scores, and AttentionSink (Xiao et al., 2023), a method that deterministically selects some initial and recent tokens. Specifically, both of these prior works have highlighted the significance of recent token embeddings in generating meaningful responses. To leverage this insight, we integrate it with our clustering approach. More precisely, our strategy consistently retains the most recent $\ell$ token embeddings, in addition to $k$ centers selected

---

[2]https://github.com/DachengLi1/LongChat/blob/longeval
[3]https://huggingface.co/lmsys/longchat-7b-v1.5-32k

| | $n = 5$k | | $n = 7$k | | $n = 9$k | |
|---|---|---|---|---|---|---|
| Algorithm | Cache Size (GB) | Accuracy | Cache Size (GB) | Accuracy | Cache Size (GB) | Accuracy |
| Exact | 2.351 | 0.98 | 3.488 | 1.0 | 4.613 | 0.68 |
| Sink (Xiao et al., 2023) | 1.511 (35% ↓) | 0.56 | 2.012 (42% ↓) | 0.56 | 2.262 (50% ↓) | 0.38 |
| H2O (Zhang et al., 2023) | 1.511 (35% ↓) | 0.66 | 2.012 (42% ↓) | 0.58 | 2.262 (50% ↓) | 0.38 |
| CLUSTERGEN | 1.512 (35% ↓) | **0.86** | 2.012 (42% ↓) | **0.66** | 2.262 (50% ↓) | **0.44** |

Table 2: Results on accuracy of line retrieval from LongEval (Li et al., 2023) dataset with context length ranging 5k to 9k. Under the sublinear cache budgets in terms of the sequence length, the proposed approach based on greedy k-center outperforms other methods over all sequence lengths.

| Algorithm | Single-QA | Multi-QA | Summurization | Fewshot | Code |
|---|---|---|---|---|---|
| Exact | 70.05 | 56.00 | 56.67 | 190.44 | 108.41 |
| Sink (Xiao et al., 2023) | 54.19 | 51.91 | 53.47 | 184.94 | 96.15 |
| CLUSTERGEN | 55.96 | 48.52 | 47.24 | 181.57 | 96.18 |

Table 3: Results on generation tasks for long-range prompts from LongBench (Li et al., 2023) datasets. The prompt length is at most 20k and the cache size budget is set to 2k, i.e., $\ell = k = 1,024$.

from the remaining tokens. In a streaming context, this strategy is often referred to as a *sliding window*. We apply the greedy k-center clustering algorithm once to compress the entire KV caches. To make comparisons fair, we set cache memory budgets of all algorithms identical (i.e., $\ell + k$), which scales sublinearly with the context length denoted as $n$. We set the compression ratio $(\ell+k)/n$ to fixed number, e.g., 0.35 for $n = 5$k, and report the highest accuracy among all combinations of $(r, k)$ where $r \in \{2048, 3072\}$ as long as $r$ does not exceed the compressed length.

The results are reported in Table 2. We observe that our clustering-based method consistently outperforms other algorithms across all sequence lengths. For instance, we achieve an accuracy of 44% while utilizing only half of the cached KV embeddings with a length of 9k tokens, whereas both H2O and AttentionSink can achieve accuracies 10% lower. This finding suggests that maintaining the embedding information holds greater significance in sustaining the performance of LLMs compared to attention scores and positional information.

### 4.2 TEXT GENERATION ON LONG-RANGE INPUTS

We evaluate our method on various tasks from LongBench (Li et al., 2023) datasets including summarization, single/multi-document question-answering, few-shot learning, and code completion. As in Section 4.1, we choose longchat-7B model and apply AttentionSink and CLUSTERGEN to the token generation process. The generated texts are evaluated using metrics from the original code Li et al. (2023). Specifically, we omit H2O because its open-source version implementation does not support memory-efficient computation for long sequences and leads the memory overflow errors. We set the maximum input length to 20,000 for all datasets and truncate the middle prompts when it overflows (i.e., first and last 10,000 tokens are appended). We fix hyperparameters $\ell, r$ to 1,024 for all datasets and both cache methods. The results are summarized in Table 2. As a result, our algorithm shows better performance scores on single-document QA and code completion tasks.

## 5 CONCLUSION

In this work, we develop CLUSTERGEN, an efficient KV cache compression algorithm via stream clustering. Our motivation is that cached keys are well clusterable in their embedding space and we apply a greedy-type clustering algorithm to find the most representative embeddings. Under assumptions on bounded query norm and clusterability, we analyze that our algorithm can guarantee a spectral error bound with sublinear time and memory. We further integrate keeping recent tokens to the proposed clustering approach. For zero-shot line retrieval tasks, our algorithm outperforms other KV cache compression algorithms with the same memory budget.

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

# A  PROOFS

## A.1  PROOF OF LEMMA 2.2

*Proof.* The first property is trivial because $\mu$ is initialized at zero and is updated in line 6 of the algorithm by adding the squared norms of $\boldsymbol{v}_i$'s. The proof of the second invariance is by induction. The base of induction holds for $n = 1$ because after processing the first token by procedure UPDATEMATRIXPRODUCT we have $\Pr[\mathcal{M}(j) = (\boldsymbol{k}_1, \boldsymbol{v}_1)] = \frac{\|\boldsymbol{v}_1\|_2^2}{\|\boldsymbol{v}_1\|_2^2} = 1$ for $j \in [s]$.

Now suppose that the inductive hypothesis holds for $n$ and we prove it must also hold for $n+1$. For any $j \in [s]$ in line 24 of Algorithm 1 with probability $p = \frac{\|\boldsymbol{v}_{n+1}\|_2^2}{\mu + \|\boldsymbol{v}_{n+1}\|_2^2}$, $\mathcal{M}(j)$ gets updated to $(\boldsymbol{k}_{n+1}, \boldsymbol{v}_{n+1})$. Since we showed that $\mu = \sum_{i \in [n]} \|\boldsymbol{v}_i\|_2^2$ we have:

$$\Pr[\mathcal{M}(j) = (\boldsymbol{k}_{n+1}, \boldsymbol{v}_{n+1})] = \frac{\|\boldsymbol{v}_{n+1}\|_2^2}{\sum_{l \in [n+1]} \|\boldsymbol{v}_l\|_2^2}.$$

Moreover with probability $1 - p = \frac{\mu}{\mu + \|\boldsymbol{v}_{n+1}\|_2^2}$, $\mathcal{M}(j)$ keeps its previous value. Using the inductive hypothesis we have that for every $i \in [n]$:

$$\Pr[\mathcal{M}(j) = (\boldsymbol{k}_i, \boldsymbol{v}_i)] = \frac{\|\boldsymbol{v}_i\|_2^2}{\sum_{l \in [n]} \|\boldsymbol{v}_l\|_2^2} \cdot \frac{\sum_{l \in [n]} \|\boldsymbol{v}_l\|_2^2}{\sum_{l \in [n+1]} \|\boldsymbol{v}_l\|_2^2} = \frac{\|\boldsymbol{v}_i\|_2^2}{\sum_{l \in [n+1]} \|\boldsymbol{v}_l\|_2^2}.$$

This completes the proof of Lemma 2.2. $\qquad\square$

## A.2 PROOF OF LEMMA 2.3

*Proof.* The proof is by induction on the stream length $n$. The base of induction trivially holds for $n = 0$, where $\mathcal{D}$ is an empty set. To prove the inductive step suppose that the inductive hypothesis holds for some $n$. Specifically, suppose that $\mathcal{D}$ is a set of $m$ items of the form $\mathcal{D} = \{(\boldsymbol{x}_i, \mathcal{S}_i, n_i) : i \in [m]\}$ and there exists a partition of keys into $m$ disjoint subsets $\mathcal{C}_1, \mathcal{C}_2, \ldots \mathcal{C}_m \subseteq \{\boldsymbol{k}_i\}_{i=1}^n$ as per in the lemma statement, such that for every $i \in [m]$: **(1)** $\boldsymbol{x}_i \in \mathcal{C}_i$, **(2)** $n_i = |\mathcal{C}_i|$, **(3)** $\|\boldsymbol{x}_i - \boldsymbol{k}'\|_2 \leq \delta$ for every $\boldsymbol{k}' \in \mathcal{C}_i$, **(4)** $\|\boldsymbol{x}_i - \boldsymbol{x}_j\|_2 > \delta$ for every $i \neq j$, and **(5)** $\mathcal{S}_i$ is a set of $t$ i.i.d. uniform samples from the set $\mathcal{C}_i$. Given this assumption, we prove that the inductive step also holds for after processing the $(n+1)$-th key in the stream $\boldsymbol{k}_{n+1}$.

In the next iteration, specifically in line 12 of UPDATESOFTMAXNORMALIZER, the algorithm finds the index $i^* \in [m]$ such that $\|\boldsymbol{x}_{i^*} - \boldsymbol{k}_{n+1}\|_2$ is minimized. Two cases arise:

**Case 1:** $\|\boldsymbol{x}_{i^*} - \boldsymbol{k}_{n+1}\|_2 \leq \delta$. In this case, the algorithm increments $n_{i^*} \leftarrow n_{i^*} + 1$ in line 14. Consider the new partitioning of the keys defined as $\mathcal{C}'_i = \mathcal{C}_i$ for $i \neq i^*$ and $\mathcal{C}'_{i^*} = \mathcal{C}_{i^*} \cup \{\boldsymbol{k}_{n+1}\}$. It follows from the inductive hypothesis that for every $i \in [m]$: **(1)** $\boldsymbol{x}_i \in \mathcal{C}'_i$, **(2)** $n_i = |\mathcal{C}'_i|$, **(3)** $\|\boldsymbol{x}_i - \boldsymbol{k}'\|_2 \leq \delta$ for every $\boldsymbol{k}' \in \mathcal{C}'_i$, and **(4)** $\|\boldsymbol{x}_i - \boldsymbol{x}_j\|_2 > \delta$ for every $i \neq j$ hold after the $n+1$-th iteration. Furthermore, since the algorithm does not alter the lists $\mathcal{S}_i$ for $i \neq i^*$, we have that **(5)** $\mathcal{S}_i$ is a set of $t$ i.i.d. uniform samples from the set $\mathcal{C}'_i$ for any $i \neq i^*$. On the other hand, the algorithm in line 17 performs reservoir sampling on the set $\mathcal{S}_{i^*}$ with new element $\boldsymbol{k}_{n+1}$ which implies that $\mathcal{S}_{i^*}$ is a set of $t$ i.i.d. uniform samples from the set $\mathcal{C}'_{i^*}$. This completes the inductive step in the first case.

**Case 2:** $\|\boldsymbol{x}_{i^*} - \boldsymbol{k}_{n+1}\|_2 > \delta$. In this case, the algorithm adds a new element to $\mathcal{D}$, thus, the updated set is $\mathcal{D}' = \{(\boldsymbol{x}_i, \mathcal{S}_i, n_i) : i \in [m+1]\}$ with $\boldsymbol{x}_{m+1} = \boldsymbol{k}_{n+1}$ and $n_{m+1} = 1$. If we consider the new partitioning of keys to be $\mathcal{C}_1, \mathcal{C}_2, \ldots \mathcal{C}_m, \mathcal{C}_{m+1}$, where $\mathcal{C}_{m+1} = \{\boldsymbol{k}_{n+1}\}$, we can use the inductive hypothesis to deduce that for any $i \in [m+1]$: **(1)** $\boldsymbol{x}_i \in \mathcal{C}_i$, **(2)** $n_i = |\mathcal{C}_i|$, **(3)** $\|\boldsymbol{x}_i - \boldsymbol{k}'\|_2 \leq \delta$ for every $\boldsymbol{k}' \in \mathcal{C}_i$, and **(4)** $\|\boldsymbol{x}_i - \boldsymbol{x}_j\|_2 > \delta$ for every $i \neq j$ hold after the $n+1$-th iteration of the stream. Furthermore, $\mathcal{S}_{m+1}$ is defined to be a list of $t$ copies of $\boldsymbol{k}_{n+1}$, thus, **(5)** $\mathcal{S}_i$ is a set of $t$ i.i.d. uniform samples from the set $\mathcal{C}_i$ for any $i \in [m+1]$. This completes the inductive step in this case and also concludes the proof of Lemma 2.3. $\qquad\square$

## A.3 PROOF OF THEOREM 2.4

*Proof.* We start the correctness proof by observing that all preconditions of Lemma 2.3 are satisfied, allowing us to invoke this lemma. Let the partition of keys into disjoint subsets be denoted by $\mathcal{C}_1, \mathcal{C}_2, \ldots \mathcal{C}_{m'} \subseteq \{\boldsymbol{k}_i\}_{i=1}^n$ satisfying $\bigcup_{j=1}^{m'} \mathcal{C}_j = \{\boldsymbol{k}_i\}_{i=1}^n$ and $\mathcal{C}_i \cap \mathcal{C}_j = \emptyset$ for every $i \neq j$ as per Lemma 2.3 for some positive integer $m'$. Rewriting the partition function in the attention denominator gives:

$$\sum_{j \in [n]} \exp(\langle \boldsymbol{k}_j, \boldsymbol{q}_n \rangle) = \sum_{i \in [m']} \sum_{\boldsymbol{k}' \in \mathcal{C}_i} \exp(\langle \boldsymbol{k}', \boldsymbol{q}_n \rangle).$$

Now by property **(3)** in Lemma 2.3 and triangle inequality, for every $i \in [m']$ and every $\boldsymbol{k}', \boldsymbol{k}'' \in \mathcal{C}_i$ we have:

$$\|\boldsymbol{k}' - \boldsymbol{k}''\|_2 \leq \|\boldsymbol{k}' - \boldsymbol{x}_i\|_2 + \|\boldsymbol{k}'' - \boldsymbol{x}_i\|_2 \leq 2\delta.$$

Therefore, using the precondition of the theorem on $\|\boldsymbol{q}_n\|_2 \leq r$ we have

$$\exp(\langle \boldsymbol{k}', \boldsymbol{q}_n \rangle) / \exp(\langle \boldsymbol{k}'', \boldsymbol{q}_n \rangle) \leq e^{2\delta \cdot r}.$$

Using the above inequality and the assumption in the theorem statement regarding $t = \Omega\left(\varepsilon^{-2} \cdot e^{2\delta \cdot r} \log n\right)$ combined with the properties **(2)** and **(5)** proved in Lemma 2.3, we can invoke Chernoff-Hoeffding inequality (see e.g., McDiarmid (1998)) along with union bound to conclude that the following holds simultaneously for all $i \in [m']$ with probability at least $1 - \frac{1}{\text{poly}(n)}$:

$$\frac{n_i}{t} \cdot \sum_{\boldsymbol{k}' \in \mathcal{S}_i} \exp(\langle \boldsymbol{q}_n, \boldsymbol{k}' \rangle) \in (1 \pm \varepsilon/3) \cdot \sum_{\boldsymbol{k}' \in \mathcal{C}_i} \exp(\langle \boldsymbol{k}', \boldsymbol{q}_n \rangle)$$

Since the terms above are positive, by summing up the given inequality for all $i \in [m']$, we find that the quantity $\tau$ computed in line 27 of Algorithm 1 satisfies the following:

$$\Pr\left[\tau \in (1 \pm \varepsilon/3) \sum_{j \in [n]} \exp(\langle \boldsymbol{k}_j, \boldsymbol{q}_n \rangle)\right] \geq 0.995 \tag{5}$$

Next, we invoke Lemma 2.2 to derive an error bound on the approximate matrix-vector product between the softmax vector and the matrix of values $\boldsymbol{V}_n$. By leveraging well-established techniques in approximate matrix products, such as the standard result from Drineas & Kannan (2001), and using the conclusion of Lemma 2.2 regarding $\mathcal{M}$ as a list of $s = \Omega(\varepsilon^{-2} \cdot d)$ i.i.d. sample from the probability distribution $\Pr[\mathcal{M}(j) = (\boldsymbol{k}_i, \boldsymbol{v}_i)] = \frac{\|\boldsymbol{v}_i\|_2^2}{\sum_{l \in [n]} \|\boldsymbol{v}_l\|_2^2}$ for $i \in [n]$ for $i \in [n]$ and $j \in [s]$, we have that vector $\boldsymbol{z}$ computed in line 26 of Algorithm 1 satisfies the following inequality with a probability of at least 0.995:

$$\left\|\boldsymbol{z} - \exp(\boldsymbol{K}_n \cdot \boldsymbol{q}_n)^\top \cdot \boldsymbol{V}_n\right\|_2$$
$$\leq \frac{\varepsilon}{3} \left\|\exp(\boldsymbol{K}_n \cdot \boldsymbol{q}_n)\right\|_2 \|\boldsymbol{V}_n\|_{op} \tag{6}$$

Now by combining inequalities in Eq. (5) and Eq. (6) using union bound and triangle inequality we find that the output of Algorithm 1 computed in line 28 as $\boldsymbol{z}/\tau$ satisfies the following with probability at least 0.99

$$\left\|\boldsymbol{z}/\tau - \texttt{softmax}(\boldsymbol{K}_n \cdot \boldsymbol{q}_n)^\top \cdot \boldsymbol{V}_n\right\|_2$$
$$\leq \varepsilon \left\|\texttt{softmax}(\boldsymbol{K}_n \cdot \boldsymbol{q}_n)\right\|_2 \|\boldsymbol{V}_n\|_{op}.$$

This completes the correctness proof of Theorem 2.4. $\qquad\square$

# B  ADDITIONAL EXPERIMENTS

## B.1  CLUSTERABILITY

We additionally provide t-SNE plots of key (first rows) and value (second rows) with more diverse layers and heads, and similar results discussed in Section 3.1 are observed; a higher degree of clusterability on the key embeddings compared to value ones.

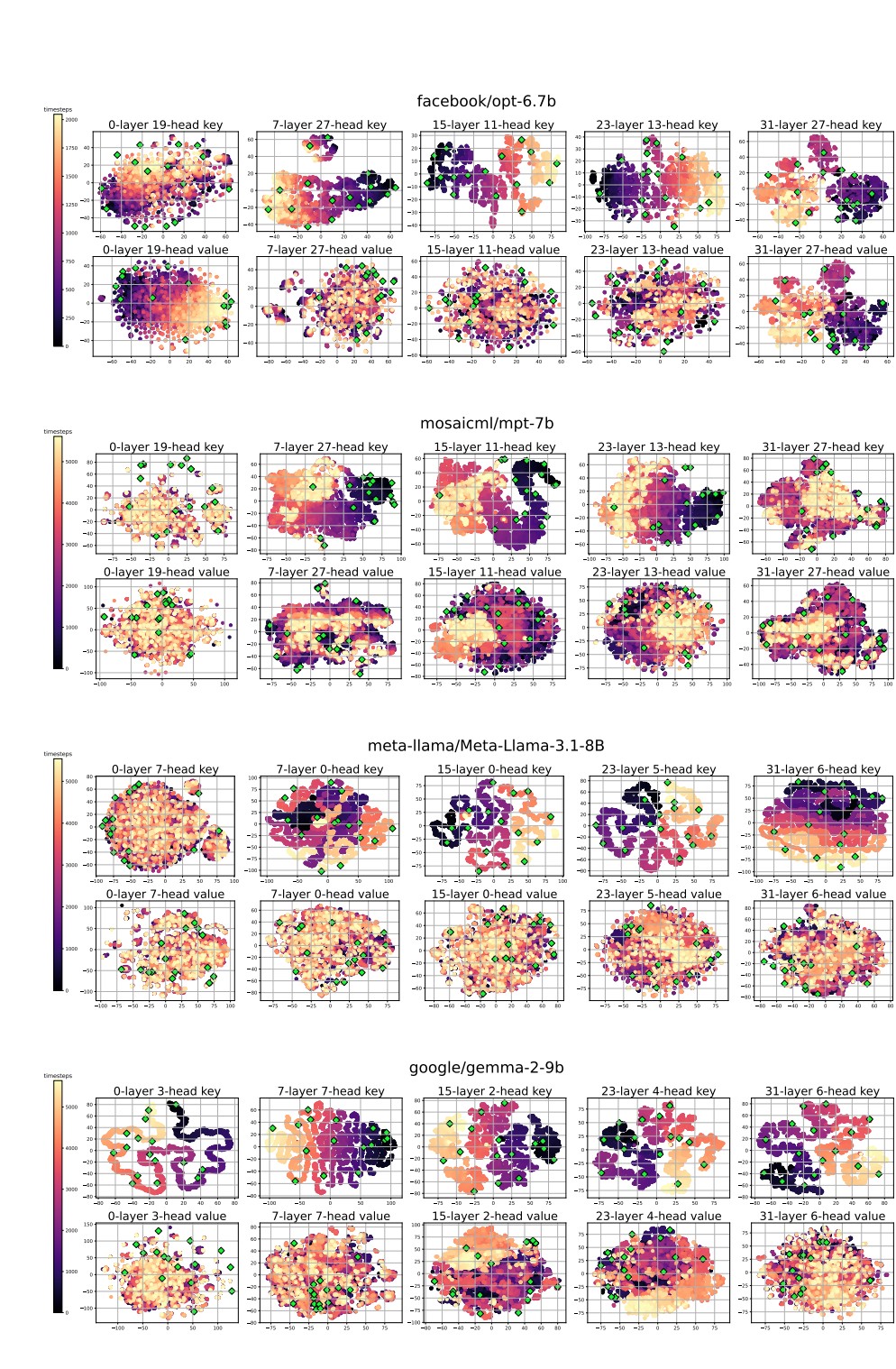

