# OpenReview forum: "ClusterGen: Token Generation in Sublinear Time and Memory with Clustering KV Cache"
_ICLR.cc/2025/Conference — Submitted to ICLR 2025_

### Official Review · Reviewer_EQco · 2024-10-22

**Soundness:** 3
**Presentation:** 3
**Contribution:** 3
**Rating:** 8
**Confidence:** 4

**Summary:**

This paper studies the problem of streaming attention problem, where key-value-query pairs are streamed to the algorithm and one needs to output the approximate attention module given by the accumulated key-value pairs and the current query. The goal is to design a data structure that uses sublinear in $n$ space. This paper achieves this goal by observing that in practice, the encodings of keys are usually well-clustered, hence it assumes that the keys can be clustered into sublinear in $n$ clusters. The data structure then consists of two main components: a dynamic sampler that approximates the matrix-matrix product, where the sampling is cleverly performed on the accumulated value matrix hence avoiding the hassle of handling the matrix involving query and entrywise exponentiation. The other component handles the normalization factor defined as $\sum_{i\in [n]} \exp(\langle k_i, q_n\rangle)$, where here it crucially uses the fact that keys are clusterable hence one could maintain a sublinear number of clusters and some uniform samples of each cluster, and approximate the normalization factor using this small coreset. Experiments are performed showcasing the effectiveness of the proposed algorithm. In particular, it was empirically checked the query vector has $O(1)$ norm and keys are generally clusterable, this enables the proposed algorithm to outperform several currently popular KV cache algorithms.

**Strengths:**

This paper studies an interesting problem that holds great theoretical and practical importance, namely developing a KV cache algorithm with sublinear memory. In practice, even an algorithm that uses constant factor smaller memory than $n$ would be valuable. The assumption imposed by this paper is quite natural, as embeddings for keys are usually clusterable among other nice properties. The proposed algorithm is simple, which is a big advantage, as it enables practical implementation. Sampling, in general could be computed quickly.

Overall, I like this paper as it studies a very practical problem from theoretical lens, obtaining good theoretical guarantees and simple algorithms, which in turn would lead to good practical solutions.

**Weaknesses:**

At least from experiments, it seems that the algorithm performs better when $n$ is smaller (5000, 7000 and 9000) compared to when $n$ is larger (20000). On the other hand, the main merit of this algorithm is when $n$ is very large as that is precisely the setting when $o(n)$ memory and update, query time is invaluable. I encourage authors to conduct more experiments on large $n$, though I do understand the difficulty as large $n$ requires more memory and compute, so this could be taken as a minor suggestion rather than a major complain.

**Questions:**

Typo:

Line 196: "probabilities proportional to $\\|k_i\\|_2^2$" should be $\\|v_i\\|_2^2$ as the importance sampling is performed on $V_n$.

Question:

Can you provide a practical guideline on how to choose the hyperparameters of the data structure?

Comment:

Throughout the paper, the use of k is quite inconsistent, sometimes it's used as k and sometimes $k$. It's better to make it consistent.

---

> ### Author Response · Authors · 2024-11-17
> **Response to Review.**
>
> Thank you for your valuable feedback, thoughtful suggestions, and you interest in our result.
>
> `Typos and Notational Consistencies:` We appreciate you pointing out the typos and inconsistencies in the use of k. We will correct these in the final version to ensure consistency throughout the paper.
>
> `Practical Guideline:` We will include a practical guide for choosing the hyperparameters in the experiments section. Specifically, our main hyperparameter is the number of clusters, which we fixed at 1024 for the long-context experiments presented in Table 3.
>
> `Additional Experiments:` We acknowledge the importance of testing with larger n. While computational constraints make this challenging, we will prioritize conducting additional experiments with longer contexts and include the results in the final version.
>
> Thank you again for your constructive feedback!

---

> > ### Comment · Reviewer_EQco · 2024-11-26
> >
> > I thank authors for the response, I'll keep my score as is.

---

### Official Review · Reviewer_BPYG · 2024-11-01

**Soundness:** 3
**Presentation:** 4
**Contribution:** 3
**Rating:** 6
**Confidence:** 4

**Summary:**

This paper gives a sublinear time and space algorithm for a streaming version of attention where query, key, and value tokens are given online and the attention needs to be approximately maintained. Some recent papers have studied using improved caching algorithms to efficiently solve this problem, and this paper aims to improve on them.

It makes a key new assumption, that the keys in the attention problem are clusterable, i.e., they can be grouped into a sublinear number of clusters which have small Euclidean radius. It verifies experimentally that RoPE-based open source models seem to satisfy this. It then gives a provably correct and efficient algorithm given this assumption. Finally, it shows via experiments that the new algorithm improves in accuracy while maintaining low memory on some tasks compared to prior work.

**Strengths:**

- The paper introduces a new clusterability assumption and backs it up with experimental evidence. I think it's particularly interesting that this assumption holds for better-performing open-source LLMs but not worse-performing ones. Focusing on this new assumption may help to guide future work in this area.
- The paper brings in the powerful toolbox of streaming clustering algorithms to give an improved attention algorithm. It's fantastic when algorithms that have been developed over decades in other contexts can be applied to modern machine learning like this.
- The new algorithm has clean, provable guarantees for accuracy, running time, and memory use.
- The paper is solving a fundamental problem for LLMs.
- The paper is generally written well.

**Weaknesses:**

- This general approach seems less effective for long-range prompts than in other settings (see table 3), but long-range prompts seem to be the main motivation for efficient token generation (see the first couple lines of section 1.1).
- There are a few details skipped, especially in the experiments section, that I'm confused about. See the questions below. In particular, the idea of using a sliding window of tokens (similar to prior work) is introduced only in the experiments section 4 but not in the algorithm theory above, and I'm unsure how much of the experimental validation is due to this and not actually the clustering.

**Questions:**

- Theorem 2.4 says the algorithm result satisfies equation (1); should it say equation (3) or (4)?
- Have sampling-based approaches for attention been used previously (in practice or theory)? I'm confused about why operator norm guarantees (eq (2)) are being used here, since they don't seem to have been used much previously for LLMs. Can you give intuition for why operator norm is a reasonable choice?
- In section 4.1 / table 2, how does the window size l compare between the three caching algorithms? Is the increase in accuracy coming because ClusterGen is able to use a bigger l? How significantly does integrating the window approach improve your algorithm? The use of the sliding window should be mentioned earlier when describing the algorithm, and not just in the experiment section.
- Did you verify the clustering assumption (definition 2.1) in the section 4 experiments? Is there a relationship between how clusterable the data is and the accuracy of the model? Does this explain why ClusterGen is better at some tasks than others in table 3?
- How did you choose which of the algorithms described in the related work (section 1.1) to compare with in the experiments (section 4)?
- How do the running times of the different algorithms compare in experiments? It seems only the memory usage is being considered.

---

> ### Author Response · Authors · 2024-11-24
> **Response to Review**
>
> Thank you for your thoughtful and detailed feedback. Below, we address the specific points raised.
>
> `Theorem 2.4 says the algorithm result satisfies equation (1); should it say equation (3) or (4)?`
>
> Theorem 2.4: Thanks for pointing out the typo. We will fix it.
>
>
> `Have sampling-based approaches for attention been used previously (in practice or theory)? I'm confused about why operator norm guarantees (eq (2)) are being used here, since they don't seem to have been used much previously for LLMs. Can you give intuition for why operator norm is a reasonable choice?`
>
> The primary focus of our paper is not the specific choice of norm guarantees but rather the development of an algorithm that leverages techniques from approximation algorithms—such as online/streaming clustering, random sampling, and approximate matrix multiplication—to address a practical problem in LLM efficiency, that is the KV cache compression. While we use operator norm guarantees (as in Eq. (3)) to establish rigorous theoretical results, the main contribution lies in applying tools from theoretical/algorithmic computer science to improve efficiency in LLMs.
>
>
> `In section 4.1 / table 2, how does the window size l compare between the three caching algorithms? Is the increase in accuracy coming because ClusterGen is able to use a bigger l? How significantly does integrating the window approach improve your algorithm? The use of the sliding window should be mentioned earlier when describing the algorithm, and not just in the experiment section.`
>
> To have a fair comparison we set the same values of the recent window size $\ell$ for all methods we compares (AttnSink, H2O, ClusterGen) in order to have equal KV cache size for all competing methods. This part is independent of those quantization methods, i.e., keeping the bottom $\ll$-by-d submatrices separately. We will update the sliding window part in the algorithm description.
>
> `Did you verify the clustering assumption (definition 2.1) in the section 4 experiments? Is there a relationship between how clusterable the data is and the accuracy of the model? Does this explain why ClusterGen is better at some tasks than others in table 3?`
>
> In **Section 3.1** and **Appendix B**, we provide empirical evidence on our clustering assumptions across different LLMs. Specifically, a better clusterability is generally observed in LLMs showing high evaluation performance based on scores of LLM leaderboard. These observations strongly support the validity and practicality of our clusterability assumption and the proposed algorithm.
>
>
> `How did you choose which of the algorithms described in the related work (section 1.1) to compare with in the experiments (section 4)?`
>
> Since our method focuses on KV cache compression through token pruning, we specifically compare it against other token pruning approaches for a fair evaluation. In our experimental section, we benchmark our ClusterGen against prominent token pruning techniques, including H2O and AttentionSink.
>
>
> `How do the running times of the different algorithms compare in experiments? It seems only the memory usage is being considered.`
>
> We evaluate the end-to-end token generation times on ``multi-news`` dataset from LongBench. Compared to the exact method (without KV cache compression), ClusterGen shows approximately  20% slower runtime in order to compress the KV cache. However, with this marginal delay, the memory space can be saved more than 5 times.

---

> > ### Comment · Reviewer_BPYG · 2024-11-26
> >
> > Thanks for the responses. My main concern remains the same: This is a nice theoretical work, but it doesn't seem to actually help with long-range prompts in the experiments (see table 3), which are the main motivation for efficient token generation (see the first couple lines of section 1.1). I'll thus leave my score as is.

---

### Official Review · Reviewer_aidT · 2024-11-03

**Soundness:** 3
**Presentation:** 3
**Contribution:** 2
**Rating:** 5
**Confidence:** 4

**Summary:**

This paper considers the problem of reducing the "KV cache" (i.e. the cache needed to compute each entry of the Attention layer). Specifically, the paper observes experimentally that the keys in the existing systems that use Attention tend to be clustered. Then it proposes to utilize this insight into creating an approximation for the Attention layer by at a _very_ high level by only only remembering the "centers" of the said clusters. Doing this reduces both the space needed to store the KV cache as well as the time needed to compute the (clustered approximation) to the Attention layer.

The paper presents algorithms/data structures to both maintain the cluster and to compute the approximation to the Attention later. Assuming $(m,\delta)$-_clusterability_ (i.e. there are $m$ clusters and the max radius of each cluster is $\delta$), the paper proves that the proposed algorithms use sub-linear time and space (as opposed to the linear time and space for the vanilla Attention implementation) (under certain assumptions on the values of $m$ and $\delta$).

The paper also presents experimental results that shows that the proposed method performs better than two existing KV cache compression systems-- Sink and H20.

**Strengths:**

* The paper presents a nice clean theoretical modeling on how Attention keys cluster (and presents some experimental results to show that this is relevant in practice).

* The paper presents nice clean theoretical bounds on the performance of the proposed algorithms (that show that they improve upon the bounds of the plain vanilla Attention implementation).

* The experimental results show improvements over two existing systems-- Sink and H20.

**Weaknesses:**

There are two major weaknesses of the paper for me-- first, the paper analyzes the improvements in time and space in the RAM/CPU model but deep learning (DL) systems and specifically Transformer models today are run on GPUs (as is the case with the experiments in the paper) and it is not clear that the theoretical improvements in the paper would translate to the GPU model. The second (and related) weakness is that the paper does not do comparison with other architectures that aim to reduce the KV cache but do have implementations that are "GPU aware."

Theoretical Analysis is on the RAM model
----------------------------------------

The theoretical analysis presented in the paper is for the RAM model and presents a random sampling based algorithm to show improvement in the RAM/streaming model. However, modern day system pretty much all run on GPUs and TPUs where random sampling does not tend to work well. This is primarily because in modern hardware all operations happen at the level of blocks-- in other words, reading one entry from a block is the same as reading the entire block. So e.g. if the random sampling does not take this block structure into account (which the algorithms in the paper from what I can tell do _not_ do), then it is not clear if the theoretical gains will manifest in actual systems since it might still be reading all the blocks even if the actual number of entries read by random sampling is much lower. Specifically, it is not clear to me that the proposed implementation would be competitive against an implementation of Attention like FlashAttention (https://github.com/Dao-AILab/flash-attention), which is an _exact_ Attention implementation that is hardware aware. Further, modern GPUs/TPUs allow for block-block matrix-multiplication in $O(1)$ time and an algorithm that is created specifically to exploit such natively supported operations might have a higher number of FLOPs than the proposed algorithm but would still beat the proposed algorithm in wall clock speed.

In summary, the modern hardware systems have native support for specialized operations, which the proposed algorithm does not seem to exploit. This does not make me confident that the proposed algorithm would work better than these hardware aware implementations.

Insufficient comparison with prior related work
-----------------------------------------------

The paper's weakness with adequate comparison with prior work can in turn be divided into three parts:

1. Of course it is entirely possible that the proposed algorithms actually run faster hardware aware algorithms (like FlashAttention) but no such comparison is provided in the experimental section of the paper. It is possible that the "Exact" Attention implementation uses a FlashAttention implementation but it is not clear from the paper whether this is the case. It would be good to either get confirmation that the "Exact" Attention uses a FlashAttention implementation (if so, which version) or if not, then it would be good to see comparison performance numbers with FlashAttention.
2. Moving beyond hardware aware exact FlashAttention implementations, there are approximate Attention implementations that have good implementations on modern hardware. Two recent examples-- The Hedgehog & the Porcupine (https://arxiv.org/abs/2402.04347) and Based (https://arxiv.org/abs/2402.18668) utilize the fact that one can approximate the exp function with low degree polynomials and then noting that apply a low degree polynomial to each entry of a low rank matrix gives rise to low rank matrices allows one to use these Taylor series approximations to define a "kernel" and then to use linear Attention instead of softmax Attention to reduce the KV cache size. These recent papers have implementations suited to modern hardware and have accuracy that matches those of Transformers (but are much more efficient).
    * I would like to point out that the above idea of using Taylor series + linear attention/low rank matrices have been exploited before. E.g. see the paper of Alman and Song (https://arxiv.org/abs/2302.13214) and the earlier work of Chen et al. (https://arxiv.org/abs/2110.15343). The latter paper also proposed to compress the KV cache under the assumption of keys being clustered (though the clustering model in this paper is cleaner and more general). But in general, for the theoretical results, it would be good to compare the theoretical results in this paper and the results e.g. in the Alman and Song paper (for the case when they get $n^{1+o(1)}$ implementation of Attention.
    * In addition to the above body of work on approximating Attention there have been alternative Attention free models that have garnered a lot of Attention. Mamba (https://arxiv.org/abs/2312.00752, also see Mamba 2: https://arxiv.org/abs/2405.21060) is probably the model that has garnered most attention. These models have comparable accuracy to Transformer but are much faster and use much smaller cache sizes.
    * Overall the paper should present a comparison with the above body of work (for approximate Attention models this should be done in both Tables 2 and 3, while for Attention free models comparison should be made in Table 3).
3. Finally, one big downside of the results presented in this paper is that the _accuracy_ over the exact Attention implementation takes a big hit. The proposed new algorithms do get an improvement of up to $50$ percent in cache size but also take a similar hit in accuracy. By comparison, the related work mentioned in the above bullet get a much larger improvement in cache size while _essentially not losing in accuracy/perplexity at all_ when compared to exact Transformers. However, these works are not evaluated in Tables 2 and 3 in the paper.

**Questions:**

Please address the two high level weaknesses pointed out above. Specifically,

1. Please comment on why the presented improvements in the RAM model would translate into wall clock improvement in modern GPU/TPUs.
2. Please compare (both the theoretical and experimental) existing results (as pointed out above and are not compared in the paper to the proposed system) with the proposed (theoretical and experimental) results.

Post-rebuttal comments
----------------------------

The authors addressed most of my concerns so am upping my score to a 5.

---

> ### Author Response · Authors · 2024-11-16
> **Response to Review.**
>
> Thank you for your thoughtful and detailed feedback. Below, we address the specific points raised.
>
> `RAM model vs GPU/TPU model:`
> We appreciate your observations regarding the applicability of our algorithm to modern hardware systems, including GPUs and TPUs. We address your concerns below, demonstrating how our algorithm can indeed be efficiently implemented in the GPU/TPU computation model and highlighting its compatibility with operations like GEMM, GEMV, and tensor reductions.
>
> 1. Line 11: Distance Computation. To compute all distances $\| x_i - k \|$ for $i \in [m]$ we can store $x_i$’s in rows of a matrix (or tensor) and compute these norms using a broadcast followed by a tensor norm reduction, which is highly optimized in current GPU libraries. Finding the index with the minimum norm is similarly efficient and can be implemented using GPU-supported argmin operations
>
> 2. Lines 15 - 16: The batched random sampling step can be done by generating a vector (tensor) of uniform random variables in the range [0,1) and comparing them against a threshold. Given that the vector dimension $t$ is a small constant in practice (e.g. 32), this operation is extremely fast. Rows passing the threshold are copied to the matrix (tensor) S, which is also a straightforward operation on GPUs.
>
> 3. Lines 24-25: Single Random Sample and Update. We sample a single random variable in the range [0,1) and compare it to a threshold. If it passes the threshold we copy the new key/value embedding vectors to the corresponding rows of the key-value cache matrix (tensor).
>
> 4. Lines 28-29: Matrix-Vector Operations. These steps can be both implemented by GEMV operations as we need to first do a matmul between the tensor of sampled key embeddings and another vector, followed by an elementwise exponential operation. Next we do an inner product with weights vector (tensor) which are inverse sampling probabilities. Then the attention scores need to be multiplied by the sampled value embeddings which can be implemented with GEMV.
>
> 5. Next note that in attention layers, computations across batch and head dimensions are entirely independent. Our algorithm naturally supports parallelism across these dimensions, ensuring it scales effectively with hardware-accelerated operations.
>
> Our algorithm in fact exploits tensor operations (e.g., GEMM/GEMV), ensuring compatibility with block-level computations on GPUs. The random sampling mechanism in our algorithm does not violate the block-level memory access paradigm, as we operate within tensorized blocks and optimize access patterns. We welcome further discussion and feedback on these implementation details.
>
>
> We address your concerns regarding experimental results in the official comment below.

---

> ### Author Response · Authors · 2024-11-16
> **Response to Review (part 2).**
>
> `Comparison with "Exact" Attention and FlashAttention implementation:`
>
> 1. There appears to be a misunderstanding about the context in which our algorithm operates. Our primary focus is on the generation phase of LLMs, where the model generates one token at a time in an online streaming mode. This phase differs significantly from the prompt phase, which involves a batch of n queries and key/value embeddings.
>
> 2. FlashAttention, as far as we know, is designed to optimize the prompt phase by leveraging techniques like checkpointing and tiling to reduce memory and computational overhead from quadratic to almost linear in n. However, during the generation phase, the runtime is less critical because attention operations occur for a single query, and the total cost scales linearly with the context size. Instead, memory concerns dominate during generation, as the KV cache size grows proportionally to the context length.
>
> 3. Our experiments explicitly target this memory bottleneck during generation. We provide comparisons focused on memory reduction, which is the critical constraint in this phase. Furthermore, in the generation phase, feed-forward layers often dominate runtime costs, while attention operations are comparatively lightweight. This justifies our emphasis on memory optimization in this context.
>
>
> `Comparison to Polynomial based methods:`
>
> Polynomial-based methods (e.g., Hedgehog, Porcupine) are an interesting line of research, but they operate under assumptions and constraints that differ significantly from ours:
>
> 1. Assumption of Small Embedding entries: These methods require key embeddings to have small entries for polynomial approximations of the exponential function to remain accurate (as shown in Alman & Song). This corresponds to a single-cluster setting in our framework. In contrast, our method supports a more general multi-cluster setting, making it applicable to more realistic scenarios where embeddings exhibit diverse structures.
>
> 2. Limitations in polynomial degree: Polynomial kernels correspond to tensor product embeddings whose dimensions grow exponentially with the degree of the polynomial. In practice, these dimensions must remain small (e.g., 3 or 4) to control the dimension of the transformed embedding vectors, seriously reducing the accuracy of the exponential approximation. Consequently, such methods require pre-training or at least fine-tuning to achieve competitive performance.
>
> 3. Post-Training Focus: Our work focuses on post-training KV cache compression, where we directly apply our method to pre-trained and fine-tuned models without requiring further training. This distinction is critical, as pretraining or fine-tuning is outside the scope of our study and we believe those problems require other types of techniques.
>
> `Comparison to alternative architectures like mamba:`
>
> - While alternative architectures such as Mamba provide intriguing pathways to efficient LLMs, they fall outside the scope of our paper. Our focus remains on dot-product attention with softmax activation and optimizing its KV cache usage during inference. Comparisons with attention-free models would require addressing architectural changes and training new models, which is a fundamentally different problem. Including such comparisons would detract from the core contributions of our work.
>
>
> `Accuracy compared to the exact Attention:`
>
> We acknowledge the trade-off between accuracy and memory savings in our method. However, it is important to reiterate that:
>
> 1. Pretraining/Fine-Tuning is Out of Scope: Unlike works that combine training-phase techniques with hardware optimizations, we focus solely on post-training inference scenarios. Naturally, fine-tuning or pretraining our method would improve accuracy but requires additional methods and resources that are not addressed in this paper.
>
> 2. Tailored to the Defined Problem: Our solution is specifically designed for post-training KV cache compression, where memory is the bottleneck during generation. Accuracy improvements via fine-tuning represent a separate problem that can build upon our work but is not its focus.

---

> > ### Comment · Reviewer_aidT · 2024-11-18
> > **Thanks for clarification on the experimental parts**
> >
> > Thanks for clarifying the point about all of this happening post-training: I quickly read through the intro and did not see this part highlighted. Stating that would be useful for the reader. Thanks for the other responses as well.
> >
> > Just as a heads up, there has been a recent work [LoLCATS](https://arxiv.org/abs/2410.10254) that also distills existing large Attention models to linear attention models (like Hedgehog/Procupine) that seems to get much better accuracy vs memory savings tradeoff. _Just to be sure, I'm **not** asking y'all to compare your work to this paper_ since that would be unfair given that the works were done independently. However, I bring this up to point out that it is possible to not lose much against plain Attention-- though _as y'all suggest_ it needs finetuning.
> >
> > However, not having any fine tuning results present in this paper, it is hard to judge if the accuracy loss with the method presented in this paper can be overcome with finetuning or not.
> >
> > Having said all of the above, the rebuttal does address my major concerns, so I'm upping my score to a 5.

---

> > > ### Author Response · Authors · 2024-11-18
> > > **Thank you for responding to our rebuttal**
> > >
> > > We sincerely thank the reviewer for their prompt response to our rebuttal and greatly appreciate their decision to raise their score.
> > >
> > > Rest assured, we will carefully address all comments and feedback to improve the presentation of our paper.
> > > We are also grateful for bringing new related research to our attention and will ensure these works are properly cited while clearly differentiating our results in relation to them.

---

> ### Comment · Reviewer_aidT · 2024-11-17
> **Thanks for the clarification on the theory part**
>
> OK, this makes sense but one thing that is not clear. If one were to count the number of block ops and GEMM/GEMV is the improvement over plain vanilla Attention would the improvement be the same? Having bounds in terms of number of block operations would be useful to see.

---

> > ### Author Response · Authors · 2024-11-18
> > **Thanks for your insightful suggestion**
> >
> > Thank you for the insightful suggestion.
> > Rest assured, we will include bounds on the number of block operations and provide additional details on the optimal implementation in the final version of our paper.

---

### Official Review · Reviewer_6U35 · 2024-11-04

**Soundness:** 3
**Presentation:** 4
**Contribution:** 3
**Rating:** 6
**Confidence:** 3

**Summary:**

The paper introduces ClusterGen, a novel caching algorithm designed to generate tokens efficiently within large language models by optimizing memory usage. It addresses the critical challenge of linear memory growth in token generation by introducing a sublinear caching technique for key-value pairs in the attention mechanism. The method leverages clustering on key embeddings and online sampling for values, achieving sublinear complexity in both time and memory. ClusterGen ensures error bounds on approximation accuracy and is empirically validated against state-of-the-art KV cache compression techniques in long-context tasks, outperforming them in both memory efficiency and processing speed​.

**Strengths:**

1. The introduction of clustering for KV cache management in LLMs is novel and distinguishing from existing methods.
2. Theoretical analyses, including complexity and error bounds, are rigorously provided.
3. The proposed ''Clusterability'' assumption is validated on real LLMs.
4. Experiments are conducted to demonstrate ClusterGen's ability as a valuable tool for practical LLM applications.

**Weaknesses:**

1. While the evaluations are robust, it could be beneficial to explore ClusterGen’s performance on additional task beyond question-answering.
2. The Clusterability assumption may not hold universally across other LLMs.

**Questions:**

1. How does ClusterGen perform on tasks other than question-answering?
2. Have the authors considered methods to handle cases with low clusterability?

---

> ### Author Response · Authors · 2024-11-24
> **Response to eview.**
>
> Thank you for your thoughtful and insightful feedback. Below, we address the specific points raised.
>
> `How does ClusterGen perform on tasks other than question-answering?`
>
> We appreciate your question about the results of other tasks. Following the reviewer’s question, we additionally experiment on a summarization task, i.e., ``multi_news’’ dataset from LongBench, comparing our method against AttentionSink. In particular, we use a fine-tuned Llama-3-8B-Instruct model [1] with context lengths ranging from 8K to over 1 million. Under the same memory constraints, our method achieves a ROUGE score (higher is better) of **27.2**, outperforming AttentionSink (**26.99**). The score of the exact method is **27.43**. This result further demonstrates our effectiveness across diverse long-context tasks. We will include a more comprehensive analysis with diverse tasks in our final draft.
>
> [1] https://huggingface.co/gradientai/Llama-3-8B-Instruct-Gradient-1048k.
>
>
> `Have the authors considered methods to handle cases with low clusterability?`
>
> We appreciate your question about low clusterability. Although we haven’t considered specific methods with low clusterability, several approaches that can be applied to the coreset selection problem (which equals to KV cache compression) would be promising future research directions. However, as we studied in Section 3.1, such clusterable properties are commonly observed in LLMs with higher evaluation scores, suggesting a correlation between the clusterability of the key cache and the evaluation performance of a given LLM on a specific dataset. We think that approaches that handle clusterability are much more important than those with low clusterability.

---

> > ### Comment · Reviewer_6U35 · 2024-11-30
> >
> > Thanks for the response. I'll keep my score as is.

---

### Meta-Review · Area_Chair_iBZT · 2024-12-15

**Metareview:**

The paper presents a new caching mechanism for better memory management of Attention mechanism. Their algorithm and improvement is based on a crucial assumption, that the keys in the attention problem are clusterable. I am not sure how universally acceptable this assumption is. Also given such clusterability assumption makes the problem inherently easy to analyze theoretically.

**Additional Comments On Reviewer Discussion:**

The rebuttal was sufficient and led to further discussion on the paper.

---

### Decision · Program_Chairs · 2025-01-22

Reject